# Postoperative fever after liver resection: Incidence, risk factors, and characteristics associated with febrile infectious complication

**Hon-Fan Lai**[1‡], **Ivy Yenwen Chau**[2‡], **Hao-Jan Lei**[1,3], **Shu-Cheng Chou**[1], **Cheng-Yuan Hsia**[1,3], **Yi-Chu Kao**[1]*, **Gar-Yang Chau**[1,3]*

**1** Division of General Surgery, Department of Surgery, Taipei Veterans General Hospital, Taipei, Taiwan, **2** Department of Otolaryngology, Cheng Hsin General Hospital, Taipei, Taiwan, **3** School of Medicine, National Yang Ming Chiao Tung University, Taipei, Taiwan

‡ HFL and IYC contributed equally to this work as first-authors.
* gychau@vghtpe.gov.tw(GYC); yckao5@vghtpe.gov.tw (YCK)

**Data Availability Statement:** All relevant data are within the manuscript and its Supporting information files.

## Abstract

### Purpose

To evaluate the incidence and risk factors of postoperative fever (POF) after liver resection. In patients with POF, predictors of febrile infectious complications were determined.

### Methods

A total of 797 consecutive patients undergoing liver resection from January 2015 to December 2019 were retrospectively investigated. POF was defined as body temperature $\geq$ 38.0°C in the postoperative period. POF was characterized by time of first fever, the highest temperature, and frequency of fever. The Institut Mutualiste Montsouris (IMM) classification was used to stratify surgical difficulty, from grade I (low), grade II (intermediate) to grade III (high). Postoperative leukocytosis was defined as a 70% increase of white blood cell count from the preoperative value. Multivariate analysis was performed to identify risk factors for POF and predictors of febrile infectious complications.

### Results

Overall, 401 patients (50.3%) developed POF. Of these, 10.5% had the time of first fever > postoperative day (POD) 2, 25.9% had fever > 38.6°C, and 60.6% had multiple fever spikes. In multivariate analysis, risk factors for POF were: IMM grade III resection (OR 1.572, p = 0.008), Charlson Comorbidity Index score > 3 (OR 1.872, p < 0.001), and serum albumin < 3.2 g/dL (OR 3.236, p = 0.023). 14.6% patients developed infectious complication, 21.9% of febrile patients and 7.1% of afebrile patients (p < 0.001). Predictors of febrile infectious complications were: fever > 38.6°C (OR 2.242, p = 0.003), time of first fever > POD2 (OR 6.002, p < 0.001), and multiple fever spikes (OR 2.039, p = 0.019). Sensitivity, specificity, positive predictive value and negative predictive value for fever > 38.6°C were 39.8%, 78.0%, 33.7% and 82.2%, respectively. A combination of fever > 38.6°C and leukocytosis provided high specificity of 95.2%.

**Funding:** GYC received a grant from Taipei Veterans General Hospital (V109C-187) https://www.vghtpe.gov.tw/, the funder had no role in study design, data collection and analysis, decision to publish, or preparation of the manuscriptndex.

**Competing interests:** The authors have declared that no competing interests exist.

## Conclusion

In this study, we found that IMM classification, CCI score, and serum albumin level related with POF development in patients undergone liver resection. Time of first fever > POD2, fever > 38.6˚C, and multiple fever spikes indicate an increased risk of febrile infectious complication. These findings may aid decision-making in patients with POF who require further diagnostic workup.

## Introduction

Postoperative fever (POF) is a common occurrence in patients undergoing major surgery, with prevalence rate ranging from 10% to 74% [1–4]. The causes of fever involve both non-infectious and infectious factors [5]. Reports of POF in patients undergoing abdominal [6, 7], spinal [8, 9], cardiovascular [10], and gynecologic surgeries [11] indicate that infection rates ranged from 5.8% to 27.0%. When evaluating POF, it is important to recognize when a wait-and-see approach is appropriate or when a further work-up is needed.

Liver resection is the standard operative treatment for liver tumors such as hepatocellular carcinoma, cholangiocarcinoma, metastatic malignancies and some benign liver diseases [12, 13]. However, there is limited concrete data on the association between fever and liver resection. In a study on postoperative antibiotic prophylaxis after liver resection, Hirokawa et al. reported that 44 of 188 patients (23.4%) had early signs of infection (defined as postoperative body temperature $\geq$ 38.0˚C and / or leukocytosis). Of those with signs of infection, 24 patients (54.5%) were diagnosed with infectious complication (including 20 surgical site infections and 10 remote site infections) [14]. Jin et al [15] described common complications related to POF, including venous catheter-related infection, pleural effusion, wound infection, pulmonary atelectasis or infection, ascites, subphrenic fluid collection or infection, and urinary tract infection.

The incidence of postoperative infectious complications in patients undergoing liver resection has been reported between 4% and 25% [16–18] and is the major cause of postoperative morbidity. Early diagnosis and treatment of postoperative infectious complications is important [19]. To our knowledge, no studies have examined the clinical significance of POF in patients undergoing liver resection. This study aims to determine the incidence and factors associated with POF after liver resection. The relationship between fever, including different patterns of fever, and infectious complication was also evaluated.

## Materials and methods

### Study population

Between January 2015 and December 2019, patients who underwent liver resection at the Department of Surgery, Taipei Veterans General Hospital, Taipei, Taiwan, were identified. Trauma, living donor hepatectomy, pediatric cases, patients with body temperature > 38˚C in the week before hepatectomy, and patients with concomitant surgeries during hepatic resection were excluded from the patients enrollment. A total of 797 cases being finally analyzed. This study was approved by the Institutional Review Board (IRB) of the Taipei Veterans General Hospital. We report a retrospective study of medical records, all data were fully anonymized before we accessed them and the IRB committee waived the requirement for informed consent.

## Variables

Body temperature was measured with a thermometer via the tympanic route, at least four times a day during the postoperative period. Postoperative fever was defined as a body temperature higher than or equal to 38.0˚C in the postoperative period. Fever was further categorized as being (1) the time of first POF ($\leq$ POD2 versus > POD2); (2) maximum body temperature ($<$ 38.6˚C versus > 38.6˚C), and (3) single versus multiple fever spikes. Infections were classified according to anatomical site (eg, surgical site, lung, urinary tract, and bloodstream). Surgical site infection was defined as a condition in which purulent discharge was observed from any incision or surgical space, with or without a positive bacterial culture. Lung, bloodstream and urinary tract infections were diagnosed based on the presence of bacteria in the discharge from the pleural cavity, sputum, blood, or urine, In some cases, the diagnosis was based on physician's judgment in patients who presented with typical signs of infection, regardless of microbiological evidence [20, 21]. The model for end-stage liver disease (MELD) score was calculated using the formula: MELD = 9.57 × $\log_e$(creatinine mg/dL) + 3.78 × $\log_e$(total bilirubin mg/dL) + 11.20 × $\log_e$(international normalized ratio, INR) + 6.43 [22]. Resection of less than three contiguous Couinaud segments was defined as minor liver resection, and resection of three or more contiguous Couinaud segments as major liver resection [23]. Liver resections were categorized into 3 levels of difficulty (low, intermediate and high) according to the Institut Mutualiste Montsouris (IMM) classification [24]. Grade I included wedge resection and left lateral sectionectomy. Grade II represented the intermediate level with anterolateral segmentectomy (IVb, V, VI, II, III) and left hepatectomy. Grade III represented the most technically advanced level including posterosuperior segmentectomy (I, IVa, VII, VIII), right posterior sectionectomy, right hepatectomy, extended right hepatectomy, central hepatectomy, and extended left hepatectomy. When multiple resections of varied difficulty were performed simultaneously, the liver resection was classified according to the most difficult procedure [25, 26]. Comorbidities before liver resection was determined using the Charlson Comorbidity Index (CCI) [27]. The CCI scores were summed for each patient and grouped into two categories: CCI score $\leq$ 3, and CCI score > 3. Pulmonary function test with measurement of forced expiratory volume in one second ($FEV_1$) was recorded. The determination of the postoperative white blood cell count was made on the basis of the value after the 4th day after liver resection [14]. Postoperative leukocytosis was defined as a 70% increase of white blood cell count from the preoperative value.

The operative procedures have been described elsewhere [28, 29]. Liver transection was performed using the tissue-fracture technique or using an energy device. Laparoscopic resection was successfully performed in 316 patients (39.6%). The techniques of inflow occlusion, either hemihepatic vascular occlusion or the Pringle maneuver, were applied [30].

The patients were carefully monitored after the surgery. A broad-spectrum prophylactic antibiotic was administered for 1~ 3 days. For patients who have clinical signs suggestive of infectious complication, a routine workup including chest radiography, sputum, drains, urine, wounds, and blood cultures was selectively arranged, depending on patients' clinical conditions. Perioperative incentive spirometry was performed to prevent atelectasis and pneumonia. Patients are discharged home when adequate mobilization, toleration of a solid diet, and pain control with oral medication are achieve.

## Statistical analysis

Variables were presented as mean (standard deviation) or number (percentage) as appropriate. The $\chi^2$ test or Fisher exact test with Yates correction was used to compare differences in categorical variables when appropriate. Continuous variables were compared using Student's t-

test. Continuous variables are dichotomized for disease risk discrimination and for decision making in clinical practice. Receiver operating characteristic (ROC) curve analysis was used to obtain optimal cutoff values for continuous variables. For (i) POF and (ii) postoperative leukocytosis as predictors of febrile infectious complication, the best cutoff values were body temperature of 38.6°C, and a 70% increase in white blood cell count from preoperative baseline, respectively. Multivariate logistic regression analysis was used to evaluate factors related to POF, and to determine risk factors related with infectious complication in patients with POF. Odds ratios (ORs) and 95% confidence intervals (CIs) were determined. Sensitivity, specificity, positive predictive value (PPV) and negative predictive value (NPV) were calculated. Statistical analyses were performed using IBM SPSS software (version 25.0; SPSS Inc., Chicago, IL). A p value < 0.05 was considered statistically significant.

## Results

During the study period, 797 patients underwent liver resection for benign or malignant diseases were included. The main characteristics of the patient were listed in Table 1. Regarding the indication for liver resection, 719 cases (90.2%) were performed for malignant diseases and 78 (9.8%) for benign diseases. Of the malignant group, 594 (82.6%) were for hepatocellular carcinoma and 64 (8.9%) for colorectal liver metastasis. In the benign group, 25 (32%) were for hemangioma and 14 (20%) for focal nodular hyperplasia.

### Prevalence of postoperative fever

Of the 797 patients, 396 (49.7%) had no fever by definition (highest body temperature <38°C during postoperative period), 138 (17.3%) had a fever of 38.0 to 38.2°C, 129 (16.2%) had a fever of 38.3 to 38.5°C, 68 (8.5%) had a fever of 38.6 to 38.8°C, 42 (5.3%) had a fever of 38.9 to 39.1°C, and 24 (3.0%) had a fever of $\geq$ 39.2°C. Overall, 401 (50.3%) of patients developed POF. The histogram of the highest postoperative body temperatures of the 797 patients is shown in Fig 1. The mean (SD) of highest body temperature was 38.0 (0.6°C, median 38.0°C, range 36.3°C to 40.6°C.

The first episode of POF occurred within POD1 for 70.6% of febrile patients (Fig 2). Fever > 38.6°C occurred in 104 patients (25.9%). Multiple fever spikes developed in 243 patients (60.6%). While there were only 20 patients (12.7%) with fever > 38.6°C in the single-spike group, 84 patients (34.6%) with multiple fever spikes had fever > 38.6°C (p < 0.001) (Fig 3).

### Risk factors for postoperative fever

In univariate analysis, six factors were found to be significantly related to the development of POF (Table 2). Postoperative fever rate increased with the IMM classification difficulty grading, from 45.2% in the grade I group, to 50.0% in the grade II group, to 57.6% in the grade III group (p = 0.003). Postoperative fever rate increased with the CCI score, from 37.3% in the 0–1 score group, to 44.6% in the 2 to 3 score group, and 59.5% in the > 3 score group (p < 0.001). Serum albumin level was another variable affecting the prevalence of POF. Patients with preoperative serum albumin level < 3.2 g/dL had 78.6% POF compared with 49.3% in the $\geq$ 3.2 g/dL group (p = 0.004). Open hepatectomy patients had a 53.4% prevalence of POF vs 45.6% in laparoscopic hepatectomy patients (p = 0.030). In patients with FEV1-predicted percentage < 95%, 54.8% of patients had POF vs 46.0% in the FEV1-predicted percentage $\geq$ 95% group (p = 0.019). Blood transfusion also had a role in POF with 55.9% of transfused patients developing fever vs 47.8% of non-transfused patients (p = 0.036)). Multivariate analysis showed that IMM classification grade III (OR 1.572, 95% CI 1.125–2.197,

**Table 1. The main characteristics of the patient sample.**

| Variables | Total (n = 797) |
|---|---|
| Age (year) | 62.1 ± 11.9 |
| Male | 553 (69.4%) |
| Body mass index (kg/m$^2$) | 24.6 ± 3.7 |
| Diabetes mellitus | 212 (26.6%) |
| Charlson Comorbidity Index | 3.27 ± 0.05 |
| Extent of hepatectomy | |
| Minor | 511 (64.1%) |
| Major | 286 (35.9%) |
| Hepatectomy | |
| Open | 481 (60.4%) |
| Laparoscopic | 316 (39.6%) |
| Repeated hepatectomy | 96 (12.0%) |
| IMM classification | |
| Grade I | 374 (46.9%) |
| Grad II | 152 (19.1%) |
| Grade III | 271 (34.0%) |
| Operative time (min) | 271 ± 104 |
| Pringle time (min) | 40 ± 25 |
| Blood loss (mL) | 899 ± 1338 |
| Blood transfusion (+) | 247 (31.0%) |
| Day 0 intake minus output (mL) | 1827 ± 1248 |
| FEV$_1$-predicted (%) | 96.9 ± 17.2 |
| Albumin (g/dL) | 3.98 ± 0.37 |
| Total bilirubin (mg/dL) | 0.82 ± 0.40 |
| Platelet, (10$^4$/μL) | 17.50 ± 7.51 |
| INR | 1.06 ± 0.07 |
| Hemoglobin, (g/dL) | 13.3 ± 1.6 |
| White blood cell (/μL) | 5632 ± 1863 |
| MELD score | 8.0 ± 2.5 |
| Length of postoperative stay | 9.83 ± 3.94 |
| Hospital costs (US dollars) | 7,784 ± 3,327 |

Values are expressed as mean ± standard deviation or the number (%) of patients

IMM classification, the Institut Mutualiste Montsouris classification; FEV1, forced expiratory volume in one second; INR, International normalized ratio; MELD, model for end-stage liver disease.

p = 0.008), CCI score > 3 (OR 1.872, 95% CI 1.325–2.645, p < 0.001), and serum albumin < 3.2 g/dL (OR 3.236, 95% CI 1.174–8.918, p = 0.023) were independent risk factors related to POF.

## Outcome of patients with postoperative fever

Of the 797 patients, 116 (14.6%) developed infectious complications Febrile complications developed in 21.9% patients with POF. There were 28 (7.1%) cases of infection in the no-fever group. Infectious complications was significantly higher in patients with POF compared to those without fever (21.9% versus 7.1%) (p < 0.001).

For patients with POF, the relationship between fever characteristic and febrile infectious complication was shown in Table 3. Patients with time of first fever > POD 2 and with

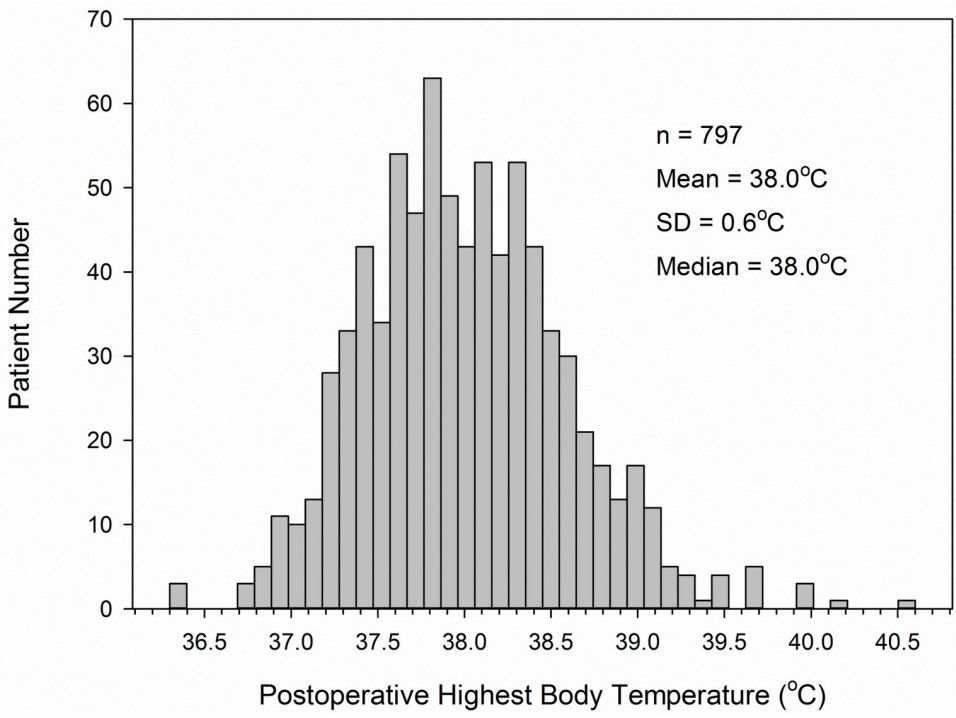

**Fig 1. Histogram of postoperative highest body temperature in 797 patients.**

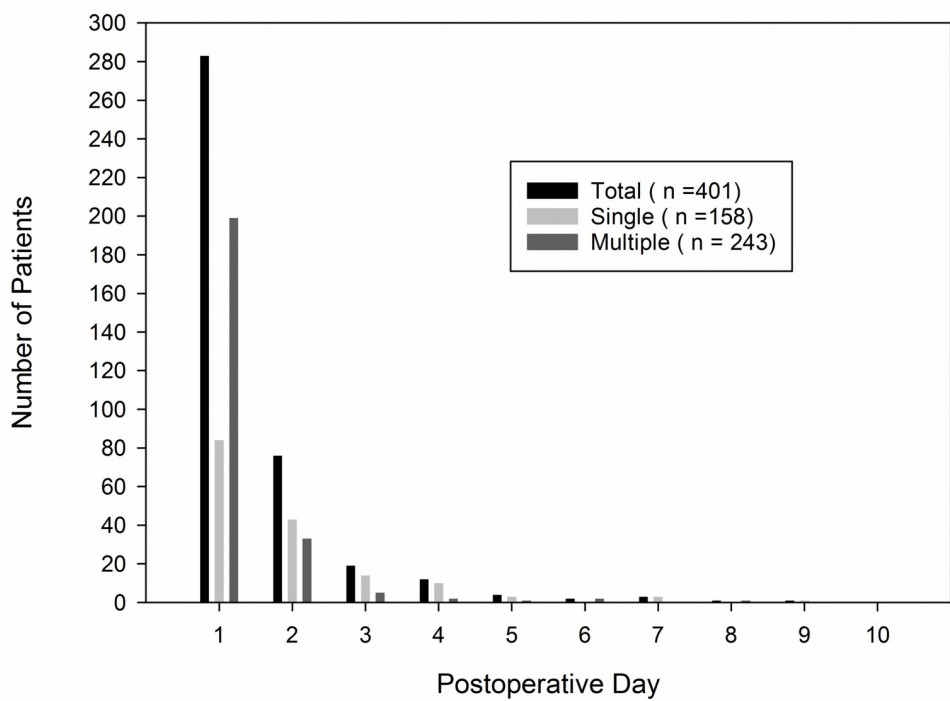

**Fig 2. The time of first fever after liver resection.** Multiple fever spike means that fever develops at least once on the different days after the first episode.

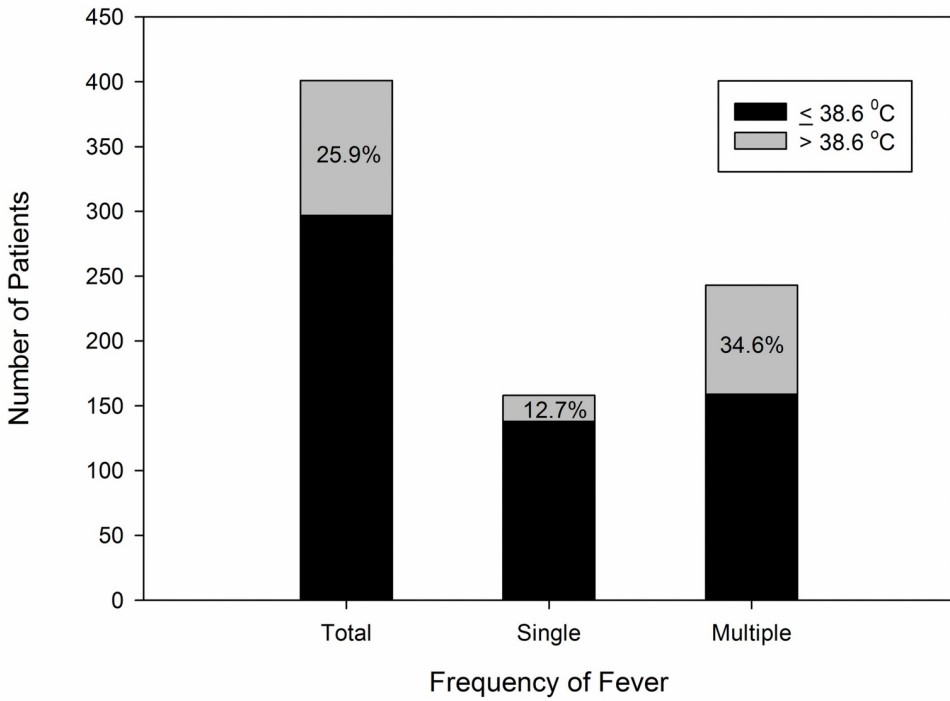

**Fig 3. The extent of maximum temperature according to the frequency of postoperative fever.** More patients had a high fever above 38.6˚C in multiple fever-spike group.

fever > 38.6˚C had significantly higher incidences of febrile infectious complications ($p < 0.001$ and $p = 0.001$, respectively) and positive bacterial cultures ($p < 0.001$ and $p = 0.001$, respectively). Patients with multiple fever spikes also tended to have increased incidence of infectious complications than those with a single fever spike, but the difference did not reach statistical significance ($p = 0.058$). The most common febrile complication was surgical site infection (47.7%). Pulmonary (including pleural cavity) infection was the second most common complication (35.2%). Blood stream infection with sepsis developed in 12.5% of all patients with POF. There was two cases of mortality (2.2%) in patients with POF, one case with postresectional liver infarction with sepsis, and one case with severe biliary tract infection and liver failure. Meanwhile, there was no cases of non-infectious febrile complications such as deep vein thrombosis or pulmonary embolism in any patients with POF.

## Length of hospital stay and hospital cost

Patients with first episode of fever after POD 2 had significantly longer postoperative hospital stay than patients with fever before POD 2 (12.5 vs 10.1 days) ($p = 0.001$). The average hospital charges were also significantly higher (US$ 10,306 vs $7,896, $p = 0.002$). Patient with POF > 38.6˚C had longer hospital stay than patients with fever ≤ 38.6˚C (11.5 vs 10.0 days, $p = 0.004$). The 30-day readmission rates were not significantly different according to the characteristics of POF (all $p > 0.05$).

## Predictors of infectious complications in febrile patients

On multivariate logistic regression analysis, fever characteristics, including fever > 38.6˚C (OR 2.242, 95% CI 1.310–3.838, $p = 0.003$), first episode of fever after POD 2 (OR 6.002, 95%

**Table 2. Risk factors for the occurrence of postoperative fever.**

| Variables | Univariate | | Multivariate | |
|---|---|---|---|---|
| | OR (95% CI) | P | OR (95% CI) | P |
| Age (> 65 vs ≤ 65) (year) | 1.305 (0.985–1.730) | 0.064 | 1.156 (0.850–1.572) | 0.355 |
| BMI (> 24 vs ≤ 24) (kg/m$^2$) | 1.129 (0.852–1.496) | 0.397 | | |
| Diabetes mellitus (yes vs no) | 1.340 (0.977–1.837) | 0.070 | 1.126 (0.772–1.642) | 0.536 |
| CCI score (> 3 vs ≤ 3) | 1.900 (1.429–2.536) | <0.001 | 1.872 (1.325–2.645) | < 0.001 |
| Resection (major vs minor) | 1.214 (0.909–1.623) | 0.189 | | |
| Resection (open vs laparoscopic) | 1.370 (1.031–1.822) | 0.030 | 1.213 (0.881–1.668) | 0.237 |
| Resection (repeated vs primary) | 1.064 (0.694–1.630) | 0.777 | | |
| IMM classification grade (III vs I/II) | 1.566 (1.158–2.091) | 0.003 | 1.572 (1.125–2.197) | 0.008 |
| Operative time (> 280 vs ≤ 280) (min) | 1.076 (0.811–1.428) | 0.612 | | |
| Pringle time (> 40 vs ≤ 40) (min) | 1.197 (0.906–1.584) | 0.206 | | |
| Blood loss (> 600 vs ≤ 600) (mL) | 1.216 (0.917–1.612) | 0.175 | | |
| Blood transfusion (yes vs no) | 1.382 (1.022–1.868) | 0.036 | 1.124 (0.798–1.584) | 0.503 |
| Day 0 intake minus output (>2000 vs ≤ 2000) (mL) | 1.230 (0.927–1.631) | 0.151 | | |
| FEV$_1$-predicted (<95 vs ≥ 95) (%) | 1.422 (1.060–1.907) | 0.019 | 1.128 (0.826–1.539) | 0.450 |
| Albumin (< 3.2 vs ≥ 3.2) (g/dL) | 3.773 (1.513–9.408) | 0.004 | 3.236 (1.174–8.918) | 0.023 |
| Total bilirubin (>1.6 vs ≤ 1.6) (mg/dL) | 1.048 (0.532–2.064) | 0.893 | | |
| Platelet (< 16.0 vs ≥16.0) (× 10$^4$/μL) | 1.183 (0.895–1.563) | 0.237 | | |
| INR (> 1.1 vs <1.1) | 1.238 (0.935–1.639) | 0.137 | | |
| Hemoglobin (< 12 vs ≥ 12) (g/dL) | 1.105 (0.774–1.579) | 0.582 | | |
| WBC count (< 3000 vs ≥ 3000) (/μL) | 1.116 (0.607–2.051) | 0.724 | | |
| MELD score (> 8 vs ≤ 8) | 1.144 (0.847–1.545) | 0.381 | | |

IMM classification, the Institut Mutualiste Montsouris classification; FEV1, forced expiratory volume in one second; INR, International normalized ratio; MELD, model for end-stage liver disease.

CI 2.849–12.643, p = 0.019), and multiple fever spikes (OR 2.039, p = 0.019) were independent predictors of febrile infectious complications (Table 4).

25.2% of febrile patients had postoperative leukocytosis. Compared with patients without leukocytosis, patients with leukocytosis had a significantly higher incidence of febrile infectious complications (37.6% vs 16.8) (p< 0.001) and positive bacterial cultures (30.7% vs

**Table 3. Outcome of the 401 patients with postoperative fever according to the characteristics of fever.**

| Variables | Extent of fever | | | Time of first fever | | | Frequency of fever | | |
|---|---|---|---|---|---|---|---|---|---|
| | 38.0~38.6˚C (n = 297) | >38.6˚C (n = 104) | P | POD ≤ 2 (n = 359) | POD > 2 (n = 42) | P | Single (n = 158) | Multiple (n = 243) | P |
| Infectious complication | 53 (17.8) | 35 (33.7) | 0.001* | 68 (18.9) | 20 (47.6) | <0.001* | 27 (17.1) | 61 (25.1) | 0.058 |
| Surgical sites | 25 | 17 | | 36 | 6 | | 11 | 31 | |
| Pulmonary | 18 | 13 | | 24 | 7 | | 11 | 20 | |
| Urinary tract | 12 | 11 | | 14 | 9 | | 10 | 13 | |
| Blood stream | 5 | 6 | | 9 | 2 | | 3 | 8 | |
| Bacterial culture (+) | 44 (14.8) | 31 (29.8) | 0.001 | 55 (15.3) | 20 (47.6) | <0.001 | 26 (16.5) | 49 (20.2) | 0.352 |
| Postoperative hospital stay (day) | 10.0±3.8 | 11.5±6.0 | 0.004 | 10.1±4.4 | 12.5±4.7 | 0.001 | 10.0±3.8 | 10.7±4.9 | 0.117 |
| Hospital cost (US dollar) | 7863 ±3709 | 8712 ±4201 | 0.078 | 7,896 ±3,430 | 10,306 ±6,750 | 0.002 | 7656 ±2459 | 8383 4525± | 0.117 |
| Readmission | 27 (9.1) | 7 (6.7) | 0.457 | 29 (8.1) | 5 (11.9) | 0.400 | 15 (9.5) | 19 (7.8) | 0.556 |

POD, postoperative day.

**Table 4. Characteristics of fever related with the occurrence of febrile infectious complication.**

| Variables | Univariate | | Multivariate | |
|---|---|---|---|---|
| | OR (95% CI) | P | OR (95% CI) | P |
| Extent of fever (>38.6 vs ≤38.6˚C) | 2.335 (1.411–3.864) | 0.001 | 2.242 (1.310–3.838) | 0.003 |
| Time of first fever (POD >2 vs ≤ 2) | 3.890 (2.010–7.532) | <0.001 | 6.002 (2.849–12.643) | 0.019 |
| Frequency (multiple vs single) | 1.626 (0.981–2.696) | 0.059 | 2.039 (1.123–3.700) | 0.019 |

POD, postoperative day.

**Table 5. Predictors of febrile infectious complications.**

| Predictors | Sensitivity, % (95% CI) | Specificity, % (95% CI) | Positive predictive value, % (95% CI) | Negative predictive value, % (95% CI) |
|---|---|---|---|---|
| Fever > 38.6˚C | 39.8 (29.6 to 50.0) | 78.0 (73.4 to 82.6) | 33.7 (24.6 to 42.8) | 82.2 (77.8 to 86.6) |
| Time of first fever (POD >2) | 22.7 (14.0 to 31.5) | 93.0 (90.2 to 95.8) | 47.6 (32.5 to 62.7) | 81.1 (77.1 to 85.1) |
| Multiple fever | 69.3 (59.7 to 78.9) | 41.9 (36.4 to 47.4) | 25.1 (19.6 to 30.6) | 82.9 (77.0 to 88.8) |
| Leukocytosis | 43.2 (32.9 to 53.5) | 79.7 (75.2 to 84.2) | 37.6 (28.2 to 74.0) | 83.2 (79.0 to 87.4) |

POD: postoperative day.

14.8%) (p < 0.001). The sensitivity, specificity, PPV, and NPV for fever characteristics and leukocytosis as predictors of febrile infectious complications are shown in Table 5. The sensitivity, specificity, PPV and NPV for fever > 38.6˚C were 39.8%, 78.0%, 33.7% and 82.2%, respectively. A combination of fever > 38.6˚C and leukocytosis provided high specificity of 95.2% (Table 6).

## Discussion

Postoperative fever is known to occur after all types of major surgical procedures with abdominal and chest procedures result in the highest incidence [6, 7, 10]. However, to date, the occurrence of fever following liver resection has not been studied. The results of the current study indicate that POF occurs in 50.3% of patients following liver resection. This incidence is slightly higher than those reported in the literature for patients undergoing major abdominal surgery, which ranged from 13% to 43% [7, 31, 32].

Fever in patients after an operation can have several causes at once, and infectious and non-infectious causes can coexist. In our study, in most patients who developed fever after liver resection, the cause is not determined. Many can be speculated to be related to tissue trauma, transfusion reaction, pulmonary atelectasis, intraabdominal fluid accumulation, transient bacteremia, or other self-limiting pathologies [5, 33, 34]. In our study, based on a large case series

**Table 6. Combined predictors of febrile infectious complications.**

| Combination of predictors | Sensitivity (95% CI) | Specificity (95% CI) | Positive predictive value, % (95% CI) | Negative predictive value, % (95% CI) |
|---|---|---|---|---|
| Fever > 38.6˚C and leukocytosis | 18.2 (10.1 to 26.3) | 95.2 (92.8 to 97.6) | 51.6 (41.1 to 62.0) | 80.5 (72.2 to 88.8) |
| Time of first fever (POD >2) and leukocytosis | 10.2 (3.9 to 16.5) | 98.1 (95.2 to 100) | 60.0 (49.8 to 70.2) | 79.5 (71.1 to 88.0) |
| Multiple fever and leukocytosis | 29.5 (20.0 to 39.0) | 89.1 (82.6 to 95.6) | 43.3 (33.0 to 53.7) | 81.8 (73.7 to 89.9) |

POD: postoperative day.

and using a multivariate analysis, we define three independent risk factors related to the development of POF. Postoperative fever was found to be more common in patients undergoing difficult types of liver resection (IMM classification grade III resection). The IMM classification is a recently reported three-level classification of the difficulty of liver resection, classifying the difficulty of the operation as low (grade I), intermediate (grade II), and high (grade III) [24]. This classification was found to be useful for stratifying surgical complexity for laparoscopic liver resection as well as open liver resection [25]. Kawaguchi et al [24] reported that the rates of major complications were highest in grade III compared to those of grade I and II. In the IMM classification, grade III resection represents the highly advanced level of surgical difficulty, including posterosuperior segmentectomy, right posterior sectionectomy, right hepatectomy, central hepatectomy, extended right hepatectomy, and extended left hepatectomy. The duration of the operation, blood loss, and morbidities differed between the three grades and gradually increased from grades I to III. The reasons why larger or complex surgery has a greater possibility of POF may be due to (1) a greater amount of tissue damage with an increased release of inflammatory cytokines, and (2) a large amount of fluid collection in a larger dead space after operation than ordinary surgery [9, 35–37]. In addition, resection of liver posterior superior segments (segments 1, 4a, 7, 8) is particularly related to the development of subphrenic fluid accumulation, pleural effusion and subsequent infectious complications [38–41]. Adequate drainage after resection in these patients is particularly important to avoid clinical symptoms associated with perihepatic fluid collection, including fever and/or abdominal discomfort [42].

The Charlson Comorbidity Index (CCI) was developed to predict the prognosis of admitted patients by assessing the number of certain comorbidities and their severity and has been widely used to assess the degree of comorbidity burden [43, 44]. The CCI system records some important comorbidity closely relevant in the context of elective liver surgery, including coexisting liver disease, chronic pulmonary disease, peptic ulcer disease, diabetes, moderate-to-severe renal disease and tumor. Ulyett et al [45] indicated that in patients undergoing liver resection, a CCI score > 4 is independently associated with the development of Clavien-Dindo grade III-V complications. Walid et al [8] has indicated that in patients undergoing spine surgery, POF rate increased with the CCI score. In our study we found that a CCI score > 3 is associated with a higher risk of development of POF.

In our study, a low level of serum albumin was found to be another factor related to the development of POF. An association between hypoalbuminemia and POF has been reported previously [46–48]. We hypothesized that serum albumin level negatively affected the POF prevalence mainly in three ways. 1) In patient undergoing liver resection and with hypoalbuminemia, fluid accumulation in the abdomen or pleural cavity is more likely to occur. 2) Studies have reported that in febrile patients, a lower serum albumin level is a predictive factor for infection. Lower albumin levels can lead to insufficient immunoglobulin synthesis, which weakens the immune system [46].; 3) Albumin has the potential to mobilize polyunsaturated fatty acids and aid in the formation of several anti-inflammatory lipids. Therefore, low levels of serum albumin may tend to pro-inflammatory status [49]. In our study, the high odds ratio of 3.23 in the prediction of POF make this a factor of high clinical relevance and easily utilized, as this data is usually known preoperatively. In patients with hypoalbuminemia, adequate replacement of albumin and fresh frozen plasma is necessary and perioperative intravenous fluids should be restricted properly [50].

Prior studies of knee, spine, and general surgery procedures have reported that fevers after POD 2, lasting longer than 24 hours, peaking above 38.9˚C, or multiple fever spikes may all be more a sign of an infectious complication [4, 37, 51–54]. There were few studies on the relationship between fever, infection, and other postoperative outcomes in patients who undergo

liver surgery. In a study of risk factors related to infections following living donor liver transplantation, Elkholy et al [55] indicated that fever was an independent predictor of early infectious complication. In our study, the incidence of infection in patients with POF was 24.5%. Three characteristics of the fever: time of first POF (POD $\leq$ 2 versus POD > 2), highest body temperature ($\leq$ 38.6˚C versus > 38.6˚C) and frequency of fever (single versus multiple) are independent predictors of febrile infectious complication (Table 4). We might be particularly aware of the possibility of infection in patients with these features, and careful evaluation should be performed for optimal control of fever. A fever workup directed by additional clinical findings for these patients can be determined individually.

In our study, analyses concerning the sensitivity, specificity, PPV, and NPV of fever characteristics as predictors of febrile infectious complications were performed (Table 5). Sensitivity and specificity, PPV and NPV for fever > 38.6˚C were 39.8%, 78.0%, 33.7% and 82.2%, respectively. Previously Vermeulen et al. [56] reported a study of 284 patients who underwent general, trauma, vascular and gastroinestinal surgery to determine the diagnostic accuracy of POF as a predictor of infectious complication. Their results revealed that a temperature $\geq$38.0˚C as cutoff point yielded a sensitivity of 37%, a specificity of 80%, a PPV of 12% and a NPV of 95%. In liver resection, postoperative leukocytosis, reflected by a >20% increase in the white blood cell count from the preoperative value, was considered an early sign of infection [14]. In our study, ROC curve analysis indicated that a 70% increase in white blood cell count from preoperative baseline is an optimal leukocytosis cut-off value as a predictor of febrile infectious complication, yielding a sensitivity of 43.2% and a specificity of 79.7%. A combination of fever > 38.6˚C with leukocytosis provided a high specificity of 95.2%. These sensitivity/specificity/PPV/NPV analyzes can further help guide the clinician and enable the application of these data.

Our study's strengths include the large sample size and included a representative population of patients undergoing liver resection, with hepatectomy performed by an experience surgical team in a recent short period of time, which adds homogeneity to this series. Nevertheless, there are limitations to our study. First, this was a retrospective study, which has inherent biases. Information on compliance with the fever management protocol is lacking. Possible missed infections may exist due to the fact that some patients who developed fever may not be worked up for a possible cause, and the determination of workup components was based on the clinical impression of the physician to the patient. Second, the course of fever may be affected by aspects of clinical management, such as antibiotic and antipyretic use. Third, it is possible that in some cases a first flare of fever occurred after discharge and was not included in the analysis. These factors were not addressed in our study, which could limit the study. However, this study provides preliminary data regarding the development of fever in patients during hospitalization after liver resection surgery that can be used as an aid to guide future research. Further prospective studies with internal or external validation are required.

## Conclusion

In this case-series study, we found that contributing factors related with POF in patients undergoing liver resection were: IMM classification grading, CCI score, and serum albumin level. In addition, first time fever after POD 2, fever > 38.6˚C, and fever with multiple spikes may indicate an increased risk of infectious complication. These parameters may be useful discriminators for potential risk of febrile infectious complication in patients undergoing liver resection with POF and may also aid decision-making in patients who require further diagnostic workup.

## Supporting information

**S1 Data.**
(SAV)

## Author Contributions

**Conceptualization:** Hon-Fan Lai, Gar-Yang Chau.

**Data curation:** Hon-Fan Lai, Ivy Yenwen Chau, Hao-Jan Lei, Shu-Cheng Chou, Cheng-Yuan Hsia, Yi-Chu Kao, Gar-Yang Chau.

**Formal analysis:** Hon-Fan Lai.

**Funding acquisition:** Gar-Yang Chau.

**Methodology:** Hon-Fan Lai, Ivy Yenwen Chau, Shu-Cheng Chou, Cheng-Yuan Hsia, Yi-Chu Kao.

**Resources:** Hao-Jan Lei.

**Writing – original draft:** Hon-Fan Lai, Ivy Yenwen Chau.

**Writing – review & editing:** Gar-Yang Chau.

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
