## [Decision Letter · Decision Letter 0]

21 Oct 2021

PONE-D-21-30756Postoperative fever after liver resection: incidence, risk factors, and characteristics associated with febrile infectious complicationPLOS ONE

Dear Dr. Chau,

Thank you for submitting your manuscript to PLOS ONE. After careful consideration, we feel that it has merit but does not fully meet PLOS ONE’s publication criteria as it currently stands. Therefore, we invite you to submit a revised version of the manuscript that addresses the points raised during the review process.

We look forward to receiving your revised manuscript.

Kind regards,

Leonidas G Koniaris, MD

Academic Editor

PLOS ONE

Journal Requirements:

3. Thank you for submitting the above manuscript to PLOS ONE. During our internal evaluation of the manuscript, we found significant text overlap between your submission and the following previously published works, some of which you are an author.

- https://www.e-neurospine.org/journal/view.php?number=796

Please revise the manuscript to rephrase the duplicated text, cite your sources, and provide details as to how the current manuscript advances on previous work. Please note that further consideration is dependent on the submission of a manuscript that addresses these concerns about the overlap in text with published work.

Reviewers' comments:

Reviewer's Responses to Questions

**Comments to the Author**

1. Is the manuscript technically sound, and do the data support the conclusions?

Reviewer #1: Yes

Reviewer #2: Yes

2. Has the statistical analysis been performed appropriately and rigorously? 

Reviewer #1: Yes

Reviewer #2: Yes

3. Have the authors made all data underlying the findings in their manuscript fully available?

Reviewer #1: Yes

Reviewer #2: Yes

4. Is the manuscript presented in an intelligible fashion and written in standard English?

Reviewer #1: Yes

Reviewer #2: Yes

5. Review Comments to the Author

Reviewer #1: This is well structured series on post-operative fever after liver resection. The data shown and the conclusions drawn show the importance of post-op fever and role for appropriate management for optimal outcomes.

Reviewer #2: Thank you for the opportunity to review this manuscript entitled “Postoperative Fever after Liver Resection: Incidence, Risk Factors, and Characteristics Associated with Febrile Infectious Complication.” The authors retrospectively reviewed 797 patients that underwent liver resection to investigate predictors of febrile infectious complications in the postoperative period (compared to non-infectious postoperative fever). 15% of patients developed infectious complications; in patients with fever, 22% developed infectious complications while 7% of afebrile patients were diagnosed with infection. Multivariable analysis associated postoperative fever with IMM Grade III resection, CCI >3, and albumin <3.2; meanwhile, predictors of febrile infectious complications included fever after POD2, Fever > 38.6 C, and multiple febrile events. The authors have put together an excellent manuscript that is a nice contribution to the literature. As the authors point out, postoperative fever after hepatectomy is frequently discussed but minimally investigated. This study could benefit from several revisions, which I have outlined in my comments below.

Major:

1. Methods – the authors elected to convert multiple continuous variables into categorical variables (for example, blood loss was categorized to either >600 or <600 mL). Can the authors clarify this decision and provide details as to how these cut off points were chosen? Was an arbitrary cut off selected? The cutoff selected will impact the results and the application of the data.

2. Methods – the authors state that febrile infectious complications were defined by positive bacterial cultures, but later in the results state that fever after POD2 and fever >38.6 were associated with febrile infectious complications AND also with positive bacterial cultures. Isn’t this just a duplicated reporting of the same result?

3. Methods/results – In clinical practice, temperature alone is rarely used to guide work-up and treatment, and clinicians combine a variety of other signs/symptoms/labs to determine if an infectious work-up is warranted (tachycardia, hypotension, leukocytosis, abdominal pain, etc). Could the authors elaborate on their analysis to include further variables, i.e. POD2 fever and leukocytosis? Multiple fevers and leukocytosis or tachycardia? Etc. Is there a certain degree of leukocytosis or elevation in temperature most strongly associated with infectious complications (analyzed by ROC curve to determine cutoff points to optimize sensitivity/specificity?)

4. Results – The authors report incidence of infectious complications among cohorts, but their study is really evaluating whether or not fever can be used to predict infectious complications. Can the authors add a more detailed analysis regarding sensitivity/specificity/PPV/NPV to help guide the clinician and allow for application of this data?

5. Discussion – nicely written discussion based on the current study and available data already published. The organization is excellent, and the discussion is thoughtful. I congratulate the authors on putting together a nice manuscript. How do the authors plan to use this information clinically, and how can readers apply this to their current practice?

Minor:

1. Introduction – I am not sure that referencing a paper from the 1970s is helpful for the introduction – could the authors find a more recent paper to demonstrate their point?

2. Methods/results – how do the authors define “borderline significance”? Most researchers would comment that a comparison is either significant (p < 0.05) or not.

6. PLOS authors have the option to publish the peer review history of their article (what does this mean?). If published, this will include your full peer review and any attached files.

Reviewer #1: No

Reviewer #2: No

---

## [Author Response · Author response to Decision Letter 0]

25 Nov 2021

Reviewer Comments, Author Responses and Manuscript Changes

Response: We have checked the manuscript to meet PLOS ONE's style requirements, including those for file naming.

2) Please provide additional details regarding participant consent. In the ethics statement in the Methods and online submission information, please ensure that you have specified (1) whether consent was informed and (2) what type you obtained (for instance, written or verbal, and if verbal, how it was documented and witnessed). If your study included minors, state whether you obtained consent from parents or guardians. If the need for consent was waived by the ethics committee, please include this information. If you are reporting a retrospective study of medical records or archived samples, please ensure that you have discussed whether all data were fully anonymized before you accessed them and/or whether the IRB or ethics committee waived the requirement for informed consent. If patients provided informed written consent to have data from their medical records used in research, please include this information.

Response: This study was approved by the Institutional Review Board (IRB) of the Taipei Veterans General Hospital. We report a retrospective study of medical records, all data were fully anonymized before we accessed them and the IRB committee waived the requirement for informed consent. (Page 5, paragraph 2). Our study did not include minors.

3) Thank you for submitting the above manuscript to PLOS ONE. During our internal evaluation of the manuscript, we found significant text overlap between your submission and the following previously published works, some of which you are an author.

Response: We apologize for this negligence and have gone through the entire manuscript carefully to avoid errors. We have revised the manuscript to rephrase duplicated text, cite the sources, and provide details as to how the current manuscript advances on previous work. 

Reviewer #1: This is well structured series on post-operative fever after liver resection. The data shown and the conclusions drawn show the importance of post-op fever and role for appropriate management for optimal outcomes.

Response: Thank you for your comments.

Reviewer #2: 

1. Methods – the authors elected to convert multiple continuous variables into categorical variables (for example, blood loss was categorized to either >600 or <600 mL). Can the authors clarify this decision and provide details as to how these cut off points were chosen? Was an arbitrary cut off selected? The cutoff selected will impact the results and the application of the data.

Response: Continuous variables are dichotomized for disease risk discrimination and for decision making in clinical practice. Receiver operating characteristic (ROC) curve analysis was used to obtain optimal cut-off values for continuous variables. (Page 8, paragraph 2).

2. Methods – the authors state that febrile infectious complications were defined by positive bacterial cultures, but later in the results state that fever after POD2 and fever >38.6 were associated with febrile infectious complications AND also with positive bacterial cultures. Isn’t this just a duplicated reporting of the same result?

Response: This is not a duplicate report of the same result. In our study, methods of confirming infection were based on bacterial culture and/or on the physician’s clinical judgment when clinical signs of infection were evident, according to guidelines issued by the Centers for Disease Control and Prevention. (Page 6, paragraph 1).

3. Methods/results – In clinical practice, temperature alone is rarely used to guide work-up and treatment, and clinicians combine a variety of other signs/symptoms/labs to determine if an infectious work-up is warranted (tachycardia, hypotension, leukocytosis, abdominal pain, etc). Could the authors elaborate on their analysis to include further variables, i.e. POD2 fever and leukocytosis? Multiple fevers and leukocytosis or tachycardia? Etc. Is there a certain degree of leukocytosis or elevation in temperature most strongly associated with infectious complications (analyzed by ROC curve to determine cutoff points to optimize sensitivity/specificity?)

Response: Receiver operating characteristic (ROC) curve analysis was used to obtain optimal cut-off values for continuous variables. The best cut-off values for (i) postoperative fever and (ii) postoperative leukocytosis as predictors of febrile infectious complication were body temperature 38.6°C, and a 70% increase of white blood cell count from the preoperative value, respectively. These points were added in Method (Page 8, paragraph 2).

4. Results – The authors report incidence of infectious complications among cohorts, but their study is really evaluating whether or not fever can be used to predict infectious complications. Can the authors add a more detailed analysis regarding sensitivity/specificity/PPV/NPV to help guide the clinician and allow for application of this data?

Response: The sensitivity, specificity, PPV and NPV of fever characteristics and leukocytosis as predictors of febrile infectious complications were listed in Table 5. (Page 19). The sensitivity, specificity, PPV and NPV of the combination of fever characteristics with leukocytosis were listed in Table 6. (Page 19).

5. Discussion – nicely written discussion based on the current study and available data already published. The organization is excellent, and the discussion is thoughtful. I congratulate the authors on putting together a nice manuscript. How do the authors plan to use this information clinically, and how can readers apply this to their current practice?

Response: Thank you for your comment. In the Conclusion, we suggest that “These parameters may be useful discriminators for potential risk of febrile infectious complication in patients undergoing liver resection with POF and may also aid decision-making in patients who require further diagnostic workup.” (Page 26, paragraph 2).

Minor:

1. Introduction – I am not sure that referencing a paper from the 1970s is helpful for the introduction – could the authors find a more recent paper to demonstrate their point?

Response: We cited a more recent paper to demonstrate our point (Reference 14: Hirokawa F, Hayashi M, Miyamoto Y, Asakuma M, Shimizu T, Komeda K, et al.. Evaluation of postoperative antibiotic prophylaxis after liver resection: a randomized controlled trial. Am J Surg. 2013; 206: 8-15.) (Page 30).

2. Methods/results – how do the authors define “borderline significance”? Most researchers would comment that a comparison is either significant (p < 0.05) or not.

Response: We changed our wording to “Patients with multiple fever spikes also had a higher incidence of infectious complications than those with a single fever spike, but the difference did not reach statistical significance (p = 0.058) (Page 16, paragraph 2).

---

## [Editor Report · Decision Letter 1]

20 Dec 2021

Postoperative fever after liver resection: incidence, risk factors, and characteristics associated with febrile infectious complication

PONE-D-21-30756R1

Dear Dr. Chau,

We’re pleased to inform you that your manuscript has been judged scientifically suitable for publication and will be formally accepted for publication once it meets all outstanding technical requirements.

Kind regards,

Leonidas G Koniaris, MD

Academic Editor

PLOS ONE
---

## [Editor Report · Acceptance letter]

5 Jan 2022

PONE-D-21-30756R1 

Postoperative fever after liver resection: incidence, risk factors, and characteristics associated with febrile infectious complication 

Dear Dr. Chau:

I'm pleased to inform you that your manuscript has been deemed suitable for publication in PLOS ONE. Congratulations! Your manuscript is now with our production department. 

Kind regards, 

on behalf of

Dr. Leonidas G Koniaris 

Academic Editor

PLOS ONE